# Asynchronous Pattern of MAPKs’ Activity during Aging of Different Tissues and of Distinct Types of Skeletal Muscle

**DOI:** 10.3390/ijms25031713

**Published:** 2024-01-30

**Authors:** Nechama Gilad, Manju Payini Mohanam, Ilona Darlyuk-Saadon, C. K. Matthew Heng, Inbar Plaschkes, Hadar Benyamini, Nikolay V. Berezhnoy, David Engelberg

**Affiliations:** 1Department of Biological Chemistry, The Institute of Life Science, The Hebrew University of Jerusalem, Jerusalem 91904, Israel; nechama.gilad@mail.huji.ac.il; 2Singapore-HUJ Alliance for Research and Enterprise, Mechanisms of Liver Inflammatory Diseases Program, National University of Singapore, Singapore 138602, Singapore; 3Department of Microbiology and Immunology, Yong Loo Lin School of Medicine, National University of Singapore, Singapore 117456, Singapore; 4Info-CORE, Bioinformatics Unit of the I-CORE, The Hebrew University of Jerusalem, Jerusalem 91120, Israel

**Keywords:** MAPKs, p38, JNK, ERK

## Abstract

The MAPK p38α was proposed to be a prominent promoter of skeletal muscle aging. The skeletal muscle tissue is composed of various muscle types, and it is not known if p38α is associated with aging in all of them. It is also not known if p38α is associated with aging of other tissues. JNK and ERK were also proposed to be associated with aging of several tissues. Nevertheless, the pattern of p38α, JNK, and ERK activity during aging was not documented. Here, we documented the levels of phosphorylated/active p38α, Erk1/2, and JNKs in several organs as well as the soleus, tibialis anterior, quadriceps, gastrocnemius, and EDL muscles of 1-, 3-, 6-, 13-, 18-, and 24-month-old mice. We report that in most tissues and skeletal muscles, the MAPKs’ activity does not change in the course of aging. In most tissues and muscles, p38α is in fact active at younger ages. The quadriceps and the lungs are exceptions, where p38α is significantly active only in mice 13 months old or older. Curiously, levels of active JNK and ERKs are also elevated in aged lungs and quadriceps. RNA-seq analysis of the quadriceps during aging revealed downregulation of proteins related to the extra-cellular matrix (ECM) and ERK signaling. A panel of mRNAs encoding cell cycle inhibitors and senescence-associated proteins, considered to be aging markers, was not found to be elevated. It seems that the pattern of MAPKs’ activation in aging, as well as expression of known ‘aging’ components, are tissue- and muscle type-specific, supporting a notion that the process of aging is tissue- and even cell-specific.

## 1. Introduction

The process of aging is a universal and a basic phenomenon that probably appeared together with life itself. Aging could be defined in various ways [1,2,3] but may simply be considered as the time passed from the organism’s birth (chronological aging), during which deleterious changes accumulate at the level of the cell, tissue, organ, and ultimately the entire organism [2]. Not only does aging itself cause deterioration of tissues and systems, it also renders the organism prone to diseases [4]. A clear example is chronic obstructive pulmonary disease (COPD), in which almost no cases are monitored among people younger than 40 years old [5,6] and which also displays hallmarks of accelerated aging [7]. Other age-associated morbidities include neurodegenerative diseases, metabolic syndrome-related diseases (including cardiovascular diseases and obesity), and muscle inflammation [8].

Skeletal muscle plays a prominent role in determining the functionality and the life quality of the aged metazoan organism. It constitutes 40% of the body mass and is responsible for 30% of the energy consumed by the entire body [9]. Aging of skeletal muscle is manifested by spontaneous atrophy, termed sarcopenia (i.e., muscle degeneration that is not a consequence of injury or denervation) [10]. Sarcopenia affects not only muscle functionality but also reduces stemness of muscle stem cells (MuSCs; also known as satellite cells) [11,12,13,14]. With defective stem cells, the tissue shows difficulties in recovery from injury and is prone to age-associated inflammation [15,16]. Understanding the aging of skeletal muscle is therefore pivotal for understanding the very phenomenon of aging. Indeed, the issue has been extensively studied, including by monitoring the entire transcriptome, proteome, and phospho-proteome during the course of aging [17,18,19,20,21,22]. Although comprehensive with respect to protein and gene expression analyses, many of these studies focused on one muscle. Bareja et al., for example, studied the quadriceps, Gannon et al. the gastrocnemius, Schaum et al. the tibialis anterior, and Lin et al. the rectus femoris [18,19,20,23]. It is not clear whether conclusions derived from studying a given muscle could be generalized. Although skeletal muscle is considered a tissue, in which many characteristics are shared by the cells that form it, different muscles possess distinct traits. At the metabolic level, for example, some types of skeletal muscles, such as the tibialis anterior and the extensor digitorum longus (EDL), mostly utilize glycolysis for ATP production and are known as “fast” muscles. They express a different set of enzymes compared to muscles like the soleus, which generate ATP primarily in the mitochondria and are known as “slow” muscles [24]. Meanwhile, other skeletal muscles, such as the gastrocnemius and the quadriceps, are known as mixed muscles [25].

The notion that aging-related observations in one tissue may not be relevant to another is further supported by large-scale analyses of gene expression [23,26,27,28,29] and suggests that aging is not uniform across tissues. While dramatic age-dependent changes in the RNA repertoire were monitored in some organs, no changes were observed in others [23,26,27,29]. In addition, within the tissue of skeletal muscles, there may be differences between cells, as observed by Shavlakadze et al., who compared changes in gene expression in several different muscles during aging. They reported that only 22% of the genes modified are shared by two or more muscles [30].

Aging of skeletal muscle, just as of other tissues, is believed to be the consequence of integrating multiple activities, suggesting that there are no aging-dedicated biochemical pathways [23]. In spite of this notion, several studies pointed at the MAP kinase p38α as a prominent aging-promoting component in muscle [14,31]. Elevated activity of this kinase in aged muscle was shown to be a major cause of reduced stemness in MuSCs [12,13,14]. Its activity promotes stem cells to undergo symmetric, rather than asymmetric, division, in which both daughter cells differentiate, depleting the MuSC pool [12,13,32,33]. The principal effect of p38α is strikingly exemplified by the observation that MuSCs from old mice are rejuvenated upon exposure to p38α inhibitors [13,14,31,32,33].

As a stress-responsive kinase that promotes cell cycle arrest and senescence, p38 might be associated with aging in other tissues as well [34,35,36,37]. Similarly, the MAP kinase JNK, which, like p38, is a stress-activated protein kinase (SAPK), was proposed to be strongly associated with aging in heart, liver, and brain [38,39,40]. Similar to the case of p38, the possible association of JNK with aging in most tissues is not known because a systematic measurement of the activities of these enzymes in different organs, specifically in distinct skeletal muscles, was not performed. A third sub-family of MAP kinases, the ERK group, could impose cell proliferation or senescence depending on the particular system in which it is activated and on the given stimulus that induced it [41,42,43,44]. It was reported to affect aging of the brain [45,46] and the immune system [47] and is also associated with aging-related metabolic diseases [48]. In skeletal muscle, its activity was found to decrease after muscle contraction in aged individuals [49]. Nonetheless, a systematic study that follows the degree of ERK’s activity in muscles and other tissues during aging has not been reported.

Here, we followed the pattern of p38, JNK, and ERK phosphorylation during aging in mouse tissues, with emphasis on the skeletal muscles. The phosphorylated forms of these kinases are the catalytically active forms [50,51,52]. It is found that the pattern of these phosphorylations is not only tissue-specific but also skeletal muscle type-specific. In some tissues and some types of skeletal muscle, these kinases are uniformly active throughout the ages measured. Of the five skeletal muscles analyzed, the quadriceps are the only muscle in which p38α phosphorylation is significantly elevated with aging, suggesting that it may not be associated with aging in most muscles.

In addition to monitoring levels of active MAPKs in muscle aging, we also followed the repertoire of muscle mRNAs during aging using RNAseq. We observed downregulation of mRNAs encoding Pax7 and upregulation of mRNAs encoding the atrophy markers Dkk3, Murf-1, and Atrogin1. However, mRNA levels of senescence markers and cell cycle inhibitors, which were previously described as aging-associated genes, did not change significantly over the course of aging. Thus, our results combined call to re-assess several proposed ‘aging markers’ and to identify tissue-specific or even cell-specific signatures of aging.

## 2. Results

### 2.1. Only Marginal Elevation of p38α Phosphorylation during Aging of the Gastrocnemius

Under the premise that p38α activity is elevated during aging in skeletal muscles [12,13,14], we set out to monitor levels of active p38α in female mice at the ages of 1, 3, 6, 13, 18, and 24 months old. Three mice were sacrificed at each age, marked A, B, and C in Figure 1, and Figures 3–12. The soleus, tibialis anterior, EDL, gastrocnemius, and quadriceps muscles were collected from each mouse, as well as the heart, brain, liver, lung, spleen, and kidney. The levels of dually phosphorylated/active p38, JNK, and ERK, as well as total levels of MK2, were monitored by Western blot using the appropriate anti-phospho antibodies.

In the gastrocnemius, Erk2 is markedly phosphorylated at all ages with no age-dependent pattern (Figure 1A lower band of row 7 and quantification in Figure 1B). Erk1 is phosphorylated at low levels and is also not affected by aging (Figure 1A upper band of row 7 and Figure 1B). p38α is phosphorylated already in young mice at ages of 1 and 3 months. Levels of phospho-p38 further increase in older mice, mainly in 18- and 24-month-old mice, to levels that are about 3 fold higher than those in 1-month-old mice, but these differences are not statistically significant below *p*-value of 0.05 (Figure 1A second row and Figure 1B). The levels of MK2, which are downregulated in some systems when p38α is highly active (reviewed in [53]), do not change during aging of the gastrocnemius (Figure 1A row 3, and Figure 1B). It is difficult to draw conclusions with respect to levels of phospho-JNK because its levels are too variable between mice (Figure 1A row 5 and Figure 1B).

### 2.2. p38γ and p38α Seem to Be Post-Translationally Modified in Muscle

As could be seen in row 2 of Figure 1, the anti-phospho-p38α antibody gives rise to two bands in the Western blot assay. One is at the apparent correct molecular weight of p38α and another, which is smeary, corresponds to a higher molecular weight (Figure 1A, row 2). What could the higher smeary band represent? As skeletal muscles express also p38γ [54], we checked whether the high molecular weight phospho-p38 proteins could include p38γ molecules. This was performed via a special Western blot assay in which the blot was probed with a mix of two primary antibodies, anti-pp38 and anti-p38γ. Next, the blot was reacted with a mix of two secondary distinct fluorescent antibodies, one reacting with the anti-phospho-p38 and the other with the anti-p38γ antibodies (Alexa fluor^®^ 488 and 568). In lysates prepared from young mice, the secondary antibody signals did not merge, suggesting that p38γ is not phosphorylated. In older mice, the antibodies did merge on a band that migrated slower than p38α and is most probably p38γ (Figure 2, upper panel; note the strong yellow color in samples of 18- and 24-month-old mice). The antibodies did not merge, however, on the smeary band above, that remained essentially green (Figure 2). Perhaps the upper, wide phospho-p38 band contains p38 molecules that were covalently modified at variable rates (hence the smeary migration) by small molecules in addition to phosphate groups (e.g., ubiquitin). To test whether p38α is modified, we performed a similar Western blot analysis as above, this time with p-p38 and p38α primary antibodies. Our results show that the p38α antibody merges only with the lower band of p-p38 (Figure 2, lower panel). A possible explanation could be that the protein underwent putative covalent modifications (i.e., phosphorylation, ubiquitination) that may prevent the recognition of the modified molecules by the anti-p38α or anti-p38γ antibodies.

### 2.3. Dramatic Differences in the Pattern of p38α, ERK, and JNK Phosphorylation between Gastrocnemius, Soleus, Tibialis Anterior, EDL, and Quadriceps over the Course of Aging

The behavior of MAPKs in the quadriceps (Figure 3) is dramatically different from that in the gastrocnemius. In this muscle, p38α’s phosphorylation is significantly elevated with aging, namely in 18 -and 24-month-old mice, as compared to the 1-month-old mice (*p* < 0.05). As is clear from Figure 3, while p38α phosphorylation is very low in 1-, 3-, and 6-month-old mice, barely detectable in some samples, it is very high in animals aged 13, 18, and 24 months (Figure 3A second row and Figure 3B). Levels in 18- and 24-month-old mice are more than 100 fold higher than those in 1- and 3-month-old mice (Figure 3B). MK2 is downregulated along with quadriceps aging (Figure 3A row 3 and Figure 3B). The behavior of Erk1/2 is similar to that of p38α. No phosphorylated Erks are detected at all in quadriceps of mice younger than 13 months old. However, in older mice, both Erk1 and Erk2 are strongly phosphorylated (Figure 3A row 7 and Figure 3B). JNK is also not phosphorylated at all in the quadriceps of young mice (1 and 3 months old), but the pattern of its activation, with respect to aging, is somewhat inconsistent (Figure 3A row 5 and Figure 3B). It seems that in the quadriceps, activity of all three MAPKs is elevated with aging.

In the soleus (Figure 4), a slow-twitch fiber type, p38α and ERK are phosphorylated in all mice tested with essentially no change over the course of aging (Figure 4A row 2 for p38, row 7 for Erk and Figure 4B). MK2 levels are also equal in all mice tested (Figure 4A row 3 and Figure 4B). JNK phosphorylation is inconsistent, causing difficulties in drawing clear conclusions. Still, its phosphorylation seems to be more elevated in mice younger than 13 months (Figure 4A row 5 and Figure 4B). In this muscle, we were able to detect the expression of histone γH2A.X, considered a marker for senescence and aging. Contrary to expectations [55,56], its levels were constant in all ages (Figure 4A row 8 and Figure 4B). In the gastrocnemius and quadriceps, γH2A.X expression could not be monitored.

In the tibialis anterior (Figure 5), a fast-twitch fiber type, p38α is phosphorylated in 1-month-old mice, but its phosphorylation is about 3-fold stronger in 3-month-old mice and remains at this level in older mice (Figure 5A row 2 and Figure 5B). MK2 levels, however, do not change (Figure 5A row 3 and Figure 5B). Curiously, while the expression levels of Erk1 are higher than those of Erk2, the phosphorylation levels of Erk2 are higher. No obvious effect of aging was observed for ERKs’ expression or phosphorylation (Figure 5A row 7 and Figure 5B). Phosphorylation of JNKs does not show a clear pattern with respect to aging, but these kinases were not phosphorylated in the tibialis anterior of 1- and 3-month-old mice (Figure 5A row 5 and Figure 5B). Finally, γH2A.X is constantly and equally expressed at all ages in the tibialis anterior (Figure 5A row 8 and Figure 5B).

In the EDL (Figure 6), p38α phosphorylation and MK2 levels are similar in most ages, with some oscillations (Figure 6A row 2 and Figure 6B). Phosphorylation of JNK1/2 seems random and does not show any correlation with aging (Figure 6A row 5 and Figure 6B). Erk1/2 behave similar to their pattern in the tibialis anterior, with high steady-state levels of Erk1 but constant phosphorylation of Erk2 (Figure 6A row 6 and Figure 6B, compared to Figure 5A,B).

In summary, the activity of MAPKs is not uniform or synchronized between skeletal muscles or even between muscles that have similar biochemical properties (fast-/slow-twitch fibers). In most of them, MAPK activity does not appear to change with aging. Only in the quadriceps, and not in any other muscle tested, their phosphorylation is clearly associated with aging.

### 2.4. Levels of Active p38α, JNK, and ERK Increase in the Course of Aging in the Lung, But Only Active Forms of JNK Increase in the Liver

Checking the status of MAPKs in other organs revealed that, similar to the observations in muscles, activity of MAPKs in aging is far from being synchronous. In the lungs, phosphorylated forms of all three MAP kinases were observed in young mice, but the levels were low. Levels of phospho-p38, phospho-ERK, and phospho-JNK increased significantly in older mice (18 or 24 months old) (*p* < 0.05) (Figure 7A,B). Levels of MK2, on the other hand, were similar at all ages (Figure 7A,B). In the heart, levels of phospho-p38 were high in mice of all ages, although somewhat higher (less than two-fold) in mice older than 6 months (Figure 8A row 2 and Figure 8B). While JNK phosphorylation does not seems to be correlated with aging in the heart (Figure 8A row 5 and Figure 8B), high levels of phosphorylated ERKs were monitored in four of the 6 mice sacrificed at the ages of 18 and 24 months old (Figure 8A row 7 and Figure 8B). In the brain, levels of phosphorylated p38α were about three-fold higher in 18- and 24-month-old mice compared to those in 1- and 3-month-old mice (Figure 9A row 2 and Figure 9B). Levels of active JNK and ERKs did not change with aging (Figure 9A,B). In the kidney (Figure 10A,B) and in the spleen (Figure 11A,B), no changes in levels of active p38, JNK, or ERKs were observed. Finally, in the liver, levels of phosphorylated JNK were elevated in mice older than 13 months, while p38 and ERKs were active at all ages (Figure 12A,B).

### 2.5. The Most Prominent Effect of Aging on the mRNA Repertoire of the Quadriceps Is the Downregulation of ECM-Related Components

Many of the effects MAPKs have on cell fate are believed to be mediated via their control on transcription initiation [57,58,59] and mRNA stability [60,61]. We thus monitored global changes in gene expression in the same mice used for monitoring MAPKs’ status. Total RNA was isolated from the quadriceps of 1-, 6, and 24-month-old mice (three mice at each age) because in this muscle, all three MAPKs were activated in the course of aging. Bulk RNA samples were analyzed by RNA-seq, and the repertoire of mRNA molecules was compared between ages. Based on this analysis, genes were grouped in six distinct clusters (Figure 13A; Appendix A lists the genes in each cluster). The results of the enrichment analysis (i.e., differentially expressed genes (DEGs); see Section 4) are shown in Figure 13B–E. Clusters 4 and 6 are composed of genes whose expression shows an age-specific pattern. Genes in cluster 4 are significantly downregulated in 24-month-old mice as compared to their levels in 1- and 6-month-old mice (Figure 13). Enrichment analysis for this cluster presents several groups of genes that belong to ECM components or regulation (Figure 13C). Also prominent in this cluster are genes associated with ERK signaling (Figure 13C). Cluster 6 includes mRNA molecules whose levels increase specifically in 24-month-old quadriceps (Figure 13A). Here, we found that most genes are classified under ‘cellular response to stimuli’ (Figure 13D). Clusters 1 and 5 are composed of genes that are similarly expressed in 1- and 24-month-old mice but expressed differently in 6-month-old mice. It is difficult to explain this observation, but it seems that these genes are associated with physiological aspects that are common to both very young and very old muscle. In cluster 1, the enrichment analysis shows several groups of genes that belong to ‘angiogenesis’, ‘ERK signaling’, and ‘ECM components’ (Figure 13B). Enriched pathways in cluster 5 belong to ‘cellular responses to stimuli’ and ‘immune system’ (Figure 13F). Clusters 2 and 3 include genes whose levels are lower (cluster 3) or higher (cluster 2) in 6- and 24-month-old mice, suggesting that they are specific to young muscle (Figure 13). In cluster 2, similar to cluster 4, most of the genes are associated with ‘ECM’ (Figure 13E) or ‘immune system’ (Figure 13E). The enrichment analysis of cluster 3 did not point at a particular group of genes.

ECM-related proteins, such as collagen and fibrinogen, are known to decrease at the mRNA level in skeletal muscle undergoing atrophy [63,64,65]. Enrichment analysis performed on the RNA-seq results from quadriceps of 1- vs. 24-month-old mice revealed that in the experiments reported here, mRNAs encoding ECM-related proteins form the group that is most significantly downregulated in the 24-month-old mice (Figure 14A; Appendix A lists the genes of each pathway). Specifically, levels of mRNAs encoding collagens (Figure 14B,C), fibronectin-1 (FN1) (Figure 14D), and fibrillin-1 (FBN1) (Figure 14E) show significant downregulation in the quadriceps of 24-month-old mice as compared to that of 1-month-old mice. The enrichment analysis revealed that not only mRNAs encoding ECM-related proteins are downregulated with aging but many other groups of genes as well. In comparison, fewer groups are upregulated (Figure 14A; Appendix A lists the genes in each pathway). A prominent upregulated pathway is the one related to adipogenesis (Figure 14A). This may be explained by the known phenomenon of adipocytes infiltration into skeletal muscle that occurs in aging [66,67].

### 2.6. Expression of Skeletal Muscle Atrophy Markers, But Not of Senescence-Associated Proteins or Cell Cycle Inhibitors, Changes in Aging Quadriceps

We next wished to monitor, over the course of aging in skeletal muscle, expression of proteins accepted as ‘aging markers’. We found among mRNAs that were upregulated those encoding the known atrophy markers Dkk3, Murf-1 (Trim63), and Atrogin-1 (Fbxo32) (Figure 15A–C). Notably, Murf-1 [68,69], Atrogin-1 [68,70], and Dkk3 [71] were not only reported to be involved in skeletal muscle atrophy but also suggested to be regulated by p38 [69,72]. In addition, downregulation of the mRNA encoding the MuSC marker, Pax7, was observed (Figure 15D). This probably reflects the decrease in the MuSCs pool. However, in contrast with previous reports, levels of mRNAs encoding the senescence markers caspase 3 and γH2AX, as well as the cell cycle inhibitor p21, which were expected to be upregulated in older mice, did not change (Figure 16A–C). Accordingly, γH2AX was detected in several tissues but showed no age-dependent changes (Figure 5A,B, Figure 6A,B, Figure 8A,B and Figure 11A,B). In summary, it seems that genes encoding muscle-specific proteins reported to be associated with aging did undergo changes in our mice, but genes reported to be associated with ‘general’ aging did not.

## 3. Discussion

MAPKs, a small sub-group within the large family of eukaryotic proteins kinases, are not only associated with most physiological processes in eukaryotes but are in fact essential for most of them [50]. It is therefore not unexpected that improper activity of these kinases is closely associated with pathologies [73,74,75,76]. As aging is also a form of morbidity, changes in the MAPKs’ activity would be anticipated in this process as well. Erk1/2 were indeed shown to be involved in aging of the immune system [47] and brain [45,77]. However, their relevance, if any, to the aging process in other tissues still awaits further investigation. The involvement of p38α/β in aging seems to be better proven, mainly in skeletal muscle, where it was shown to be a major factor in aging-associated reduced stemness of MuSCs [14,31,32,33]. However, the patterns of ERK, JNK, and p38 activities in the course of aging were hitherto not documented.

The systematic mapping of these activities reported here suggests that at least with respect to MAPKs’ activity, there is no common pattern of changes during aging across tissues, or even across cell types within the same tissue. It implies that aging of each muscle should be studied individually. These future studies may or may not identify common mechanistic denominators in the aging process for all muscles. Although it seems that p38, JNK, or ERK are not common denominators of muscle aging, this notion should be taken with caution since the analysis performed here was performed on the whole tissues, namely in low resolution. The possibility does remain that for cell types within the tissue, such as MuSCs, p38 or other components are synchronously activated during the course of aging in all skeletal muscles in the body. This should be checked via single-cell analysis in each type of muscle. We guesstimate, however, that at the single-cell level too, differences between tissues and muscles would be apparent, and a common denominator of the aging process would be difficult to find.

In some muscles (e.g., gastrocnemius), p38α’s activity does not change much in the course of aging, but there is accumulation of molecules of phosphorylated p38 that migrate slower than the regular p38α molecule in SDS-PAGE and create a smeary band. Perhaps it is the activation of p38γ combined with a currently unidentified post-translational modification (e.g., ubiquitination) that creates this phenomenon. The nature of these phosphorylated p38 molecules appears to be muscle-specific. Such smears are observed in gastrocnemius, quadriceps, and tibialis anterior and are rarely detected in other tissues (they do appear in the lungs of some mice; Figure 7 row 2).

The lung is an exception, in which all three MAPKs are activated in the course of aging. In this regard, the lung is similar to the quadriceps. The pattern of ERK and JNK activities in aging lungs was not previously reported, but p38α phosphorylation was noted in response to LPS treatment of aged mice [78]. The elevation of all MAPKs in lungs of aging mice may be related to the fact that MAPKs are more efficiently activated in an oxidative environment [79,80,81]. This notion is based on the presumption that the lungs accumulate more reactive oxygen species (ROS) during aging [82,83,84]. It would be interesting to test the ROS levels in the aged quadriceps. The lung is very prone to aging-associated disease, mostly via oxidative stress and NF-kB signaling [78,81,84], both of which are related to the activation of MAPKs. p38 was particularly shown to be involved in lung diseases such as COPD and asthma, mainly by regulating inflammation-related genes via its direct substrate MK2 [85,86].

p38 and JNK are also known as stress-activated protein kinases (SAPKs) and were reported to induce protective responses such as cell cycle arrest, chaperone activity, repair systems, and the apoptosis of unrepairable cells in response to stress [87]. Activation of ERK, depending on the cell type and physiological context, may lead to opposite responses such as rapid proliferation or senescence [57]. During the aging process, ERK activity is believed to promote senescence, one of the hallmarks of aging [43,57]. Its activation in the lung and quadriceps may also be related to the accumulation of damaged or defective cells due to increased ROS levels, arresting them prior to apoptosis [84,88].

Since we analyzed tissues as bulk samples, it is impossible to say whether ERK, JNK, and p38 are co-activated in the very same cells of the lung or the quadriceps. Important in this regard is the study of Zhang et al., who found, through single cell RNA analysis, that p16 is expressed just in a subset of aged cells and that these particular cells possess damaged DNA [22]. Interestingly, Zhang et al. also found that p21 is expressed in a different subset of cells [22]. Namely, aged cells may only show some, but not necessarily all, of the proposed aging markers.

The conclusion that emerges from the analysis of MAPKs during aging joins the conclusions of others that aging is far from being synchronous between tissues and cell types [23,27,89,90]. It could in fact be considered unique to individual cells [91]. In this regard, MAPKs seem to play a more significant role in the aging of lung, heart, brain, and quadriceps as compared to other tissues.

## 4. Materials and Methods

### 4.1. Mice

For testing the effects of aging, wild-type C57BL/6J female mice were grown at Invivos Laboratory (Singapore) to the ages of 1, 3, 6, 13, 18, and 24 months old, transferred to the animal facility of The National University of Singapore (NUS), and sacrificed 3 days later. Each age group included 3 mice. Skeletal muscles (quadriceps, gastrocnemius, tibialis anterior, EDL, and soleus), heart, lungs, liver, kidney, spleen, and brain were collected, snap-frozen on liquid nitrogen, and then stored at −80 °C.

All mice were housed in ventilated cages with a constant temperature of 25 °C, 30–70% humidity, 12-h light/dark cycle. The protocols for mouse breeding, BR20-0795, and research, R20-0799, were approved by the IACUC of NUS, and breeding and research protocol NS-19-15807-4 was approved by the IACUC of the Hebrew University of Jerusalem.

### 4.2. Anesthesia

Mice were injected intra-peritoneally with an overdose of 0.75 mg/mL Ketamine + 0.1 mg/mL Medetomidine/Saline cocktail (purchased from the animal pharmacy in MD2, NUS) 0.015 mL/gr Medetomidine followed by cardiac puncture. Tissues were then collected, processed, and/or stored for the analysis as detailed below.

### 4.3. Protein Lysate Preparations

Protein extracts of mice organs were prepared from liquid nitrogen snap-frozen organs in 2× tissue volume lysis buffer (50 mM HEPES pH = 7.5, 150 mM NaCl, 1 mM EDTA, 1% Triton X-100, 0.1% sodium deoxycholate, 0.1% SDS) that also contained HALT^TM^ Protease and Phosphatase Inhibitor Cocktail (1009) (#78440; Thermo Fisher Scientific, Waltham, MA, USA). Tissues were homogenized using a Bullet Blender Tissue Homogenizer with the relevant beads (1× tissue volume-Next Advance, Troy, NY, USA) according to the manufacturer’s instructions. Supernatant was collected and mixed with 2× SDS sample buffer and boiled for 10 min.

Protein quantification was performed with an MN Protein Quantification Assay (250) Reagents Kit (740967.250 MACHEREY-NAGAL, Duren, Germany).

### 4.4. Immunoblotting

A total of 30 μg of protein samples was separated using 12% SDS/PAGE (5% of stacking gel concentration) and transferred to a nitrocellulose membrane (Bio-Rad) using the Trans-Blot Turbo System (Bio-Rad). Membranes were blocked with 5% non-fat milk in TBST (1× PBS with 0.1% Tween) for 60 min and then incubated with a primary antibody (diluted in 5% BSA in TBST) for 15 h, washed, and incubated with an HRP-conjugated secondary antibody (diluted in 5% nonfat dry milk in TBST) for 90 min. The signal was developed using the Western BrightTM ECL Kit (K-12045-D50 Advansta, San Jose, CA USA) or Maximum Sensitivity Substrate Kit (34096 Thermo Fisher). The signal was detected using the ChemiDOC^TM^ MS Imaging System instrument (Bio-Rad Laboratories, Hercules, CA, USA). All detected proteins appeared at the expected apparent molecular weight.

**Primary antibodies used:** anti-GAPDH (Ab8245; Abcam, Cambridge, UK), anti-p38α (C20) (SC-535-G; Santa Cruz, Dallas, TX, USA), anti-p38γ (MAB1347; R&D system), anti-phos-p38 (#4511), anti-MAPKAPK-2 (MK2) (#3042), anti-JNK2 (56G8) (#9258), anti-Phospho-JNK (#4671), anti-ERK (#9102), anti-Phospho-ERK (Thr202/Tyr204) (D13.14.4E) (#4370), and anti-Phospho-Histone γH2A.X (Ser139) (#9718S) were purchased from ‘Cell Signaling technology’, (Danvers, MA, USA). **HRP-conjugated secondary antibodies used:** anti-rat (SC-2006) and anti-goat (SC-2354) were purchased from Santa Cruz, Dallas, TX, USA. Anti-rabbit (#7074) and anti-mouse (#7076) were purchased from Cell Signaling technology. **Secondary fluorescent antibodies:** Goat Anti-Mouse IgG H&L (Alexa Fluor^®^ 568) (ab175701) and Goat Anti-Rabbit IgG H&L (Alexa Fluor^®^ 488) (ab150081) were purchased from Abcam, Cambridge, UK.

### 4.5. mRNA Extraction

mRNA was extracted from snap-frozen murine skeletal muscle tissues using a Bullet Blender Tissue Homogenizer with stainless steel beads (Blend 0.9 to 2.0 mm) (Next Advance—NA SSB14B) in PureZOLTM RNA Isolation Reagent (Bio-rad, #7326890) and subsequently by Aurum^TM^ Total RNA Fatty and Fibrous Tissue Module (Bio-rad, #7326870).

### 4.6. RNA-Seq Analysis

#### 4.6.1. Detection of Differentially Expressed Genes

Raw reads were quality-trimmed, and remaining adapter sequences were removed using cutadapt [92]. Processed reads were aligned to the mouse genome version GRCm39 (with genome annotations of Ensembl release 106) with TopHat [93]. Quantification was done with htseq-count [94] with strand information set to ‘reverse’. DESeq2 analysis [95] for identification of differentially expressed genes was performed, and pair-wise comparisons were tested with default parameters, except not using the independent filtering algorithm. The significance threshold was set to *p* adj < 0.1. In addition, significant genes were further filtered by the log2FoldChange value. This filtering was baseMean-dependent and required a baseMean above 5 and an absolute log2FoldChange higher than 5/sqrt(baseMean) + 0.5 (for highly expressed genes, this means a requirement for a fold-change of at least 1.5, while genes with a very low expression would need a 7-fold change to pass the filtering).

#### 4.6.2. k-Mean Clustering of DE Genes

To test the system as a whole, a likelihood ratio test (LRT) against the intercept (~1) was utilized. This type of test gives each gene an FDR-corrected *p*-value that indicates whether the expression pattern is more likely to stem from the different conditions than from differences between the biological replicates. Genes with adjusted *p*-value < 0.05 were selected as having their expression affected by the different experimental conditions. Normalized counts per sample of those selected genes of biological replicates were averaged and standardized per gene expression vector, i.e., subtracting the mean and dividing by standard deviation (z-scored, mean 0 standard deviation 1). The standardized expression matrix was subjected to k-mean clustering with alternating k values between 2 and 8. The k = 6 solutions were selected for display. The “heatmap” R package was used for generation of heat maps of genes’ average standardized expression per treatment ordered by hierarchal clustering.

### 4.7. Enrichment Analysis

Differentially expressed genes (DEGs) from K = 6 solution (Figure 13A–E) and the comparison between 24-month-old vs. 1-month-old mice (Figure 14A) were submitted to enrichment analysis using GeneAnalytics (Ben-Ari Fuchs et al. [62]). To learn which pathways were altered mainly due to up or downregulation of genes, we used two separated lists of DEGs that displayed significant higher or lower expression in 24-month vs. 1-month-old mice. We considered only pathways with high score, corresponding to corrected *p*-value smaller than 0.0001.

### 4.8. Statistical Analysis

Statistical significance of the differences between groups was determined by ordinary one-way ANOVA or unpaired two-tailed *t*-test with the GRAPHPAD PRISM software, (v9.5.1) San Diego, CA, USA (* *p* < 0.05, ** *p* < 0.01, *** *p* < 0.001, **** *p* < 0.0001). All data are expressed as mean ± S.E.M.

## 5. Conclusions

The main conclusion of this study is that each tissue, and even each type of skeletal muscle, is unique and distinct with respect to the dynamics of MAPKs’ activities over the course of aging, emphasizing the asynchronistic nature of aging even within a single tissue. Even activity of the MAPK p38α, considered to promote aging in skeletal muscle, is constant in most muscles and is significantly elevated only in the quadriceps. Another conclusion is that aging in skeletal muscle (quadriceps) is associated mainly with downregulation rather than upregulation of gene expression. Particularly downregulated are genes encoding ECM components. Finally, the acceptable aging markers (i.e., senescence-associated proteins and cell cycle inhibitors) did not show any aging-dependent change. Instead, skeletal muscle-specific genes, known to increase during atrophy, were elevated in aging. These conclusions suggest that characteristics of aging should be considered tissue-/organ-specific, or perhaps even cell type-specific.

## Figures and Tables

**Figure 1 ijms-25-01713-f001:**
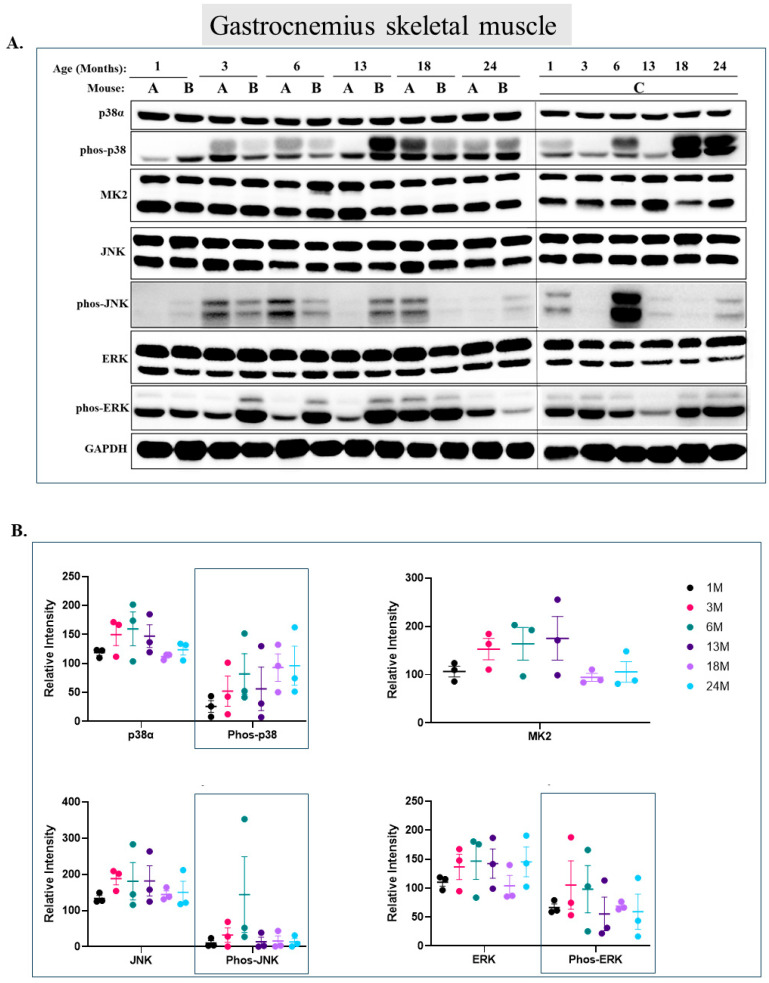
**Marginal elevation in phosphorylation of p38α, but not of ERKs and JNK, in the gastrocnemius during aging.** (**A**) Western blot analysis of protein lysates prepared from the gastrocnemius of wild-type mice, collected at the indicated age and tested with the indicated antibodies. Three mice were sacrificed at each age and were marked A, B, and C. (**B**) Relative intensity of Western blots presented in panel A, as measured by ImageJ and normalized to the levels of GAPDH. Statistical analysis revealed no significant differences.

**Figure 2 ijms-25-01713-f002:**
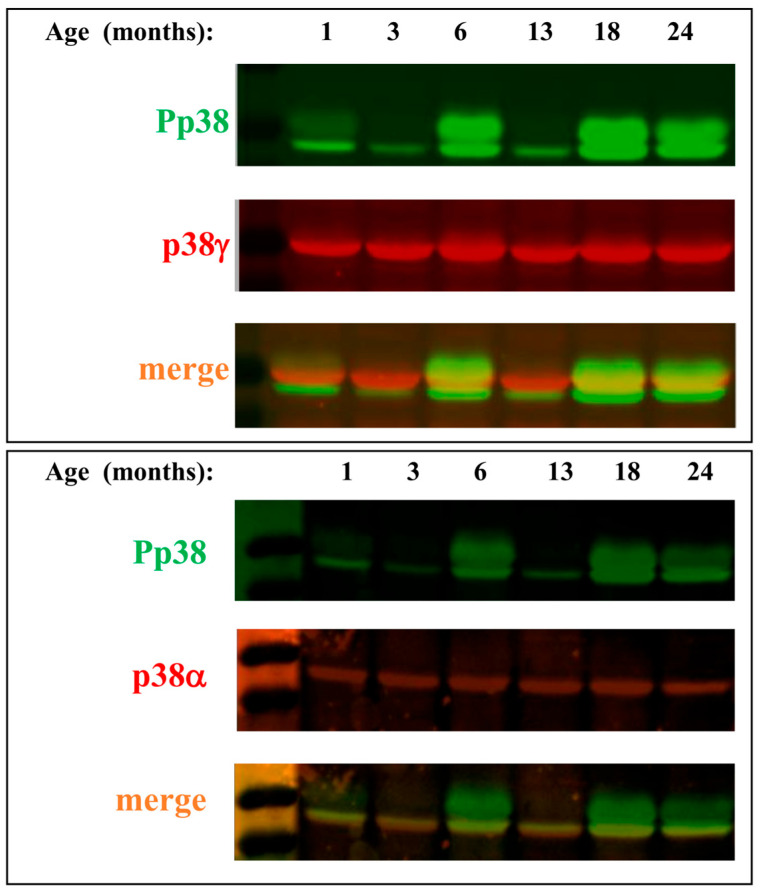
**p38γ and p38α seem to undergo massive post-translational modifications during aging of skeletal muscle.** Protein lysates prepared from gastrocnemius of the mice marked A in Figure 1A were analyzed by Western blot that was probed with a mix of anti-phos-p38 plus anti-p38γ (**upper panel**) or p38α (**lower panel**) antibodies and then with a mix of fluorescent secondary antibodies that mark p38γ or p38α in red and phosphorylated p38 in green (see details in Section 4).

**Figure 3 ijms-25-01713-f003:**
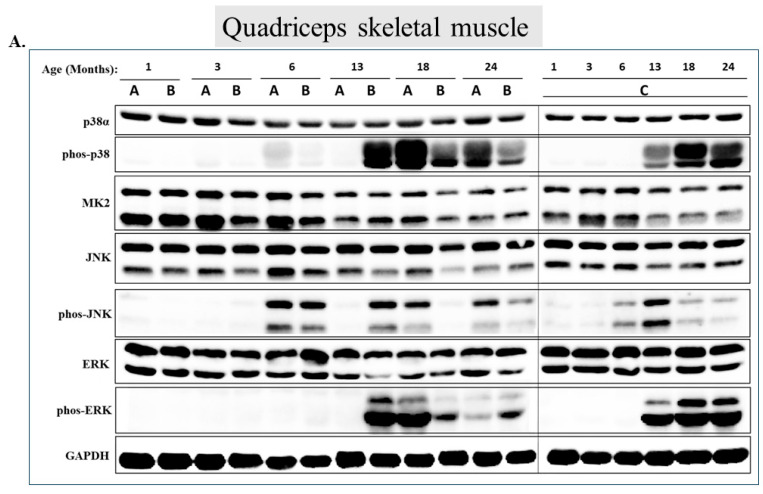
**p38α and Erk1/2 are strongly activated in the quadriceps during aging.** (**A**) Western blot analysis of protein lysates prepared from the quadriceps of wild-type mice, collected at the indicated age and tested with the indicated antibodies. Three mice were sacrificed at each age and were marked A, B, and C. (**B**) Relative intensity of Western blot results presented in panel A, as measured by ImageJ and normalized to the levels of GAPDH. Statistical significance of the differences of expression levels between 1- and 24-month-old and between 1- and 18-month-old mice was determined by an unpaired two-tailed *t*-test using GRAPHPAD PRISM software (v9.5.1) (ns = non-significant. Asterisks (*) *p*-value < 0.05).

**Figure 4 ijms-25-01713-f004:**
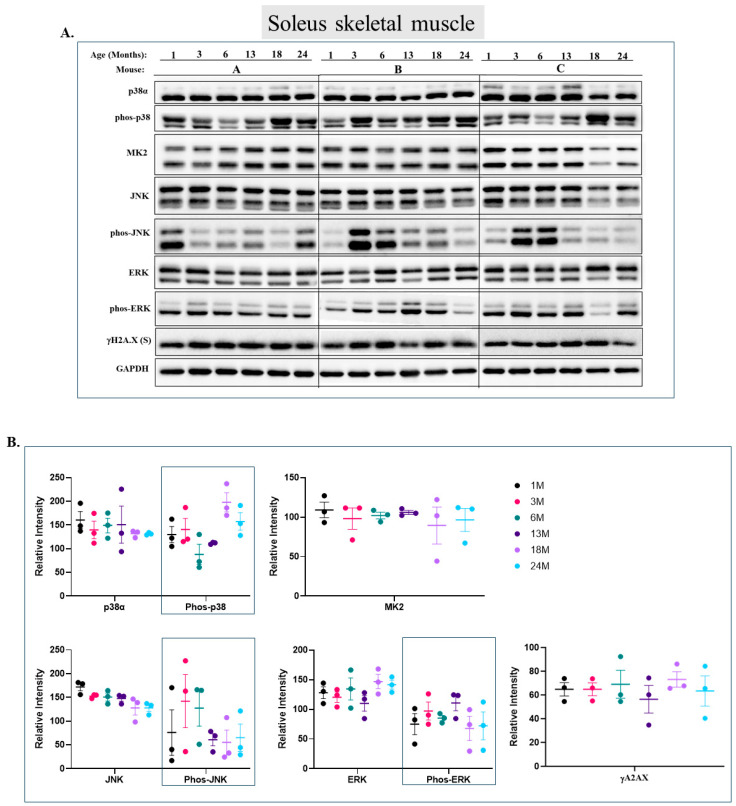
**Phosphorylation levels of p38α, ERKs, and JNK are essentially constant in the soleus during aging.** (**A**) Western blot analysis of protein lysates prepared from soleus skeletal muscle of wild-type mice, collected at the indicated age and tested with the indicated antibodies. Three mice were sacrificed at each age and were marked A, B, and C. (**B**) Relative intensity of Western blot results presented in panel A, as measured by ImageJ and normalized to the levels of GAPDH. Statistical analysis revealed no significant differences.

**Figure 5 ijms-25-01713-f005:**
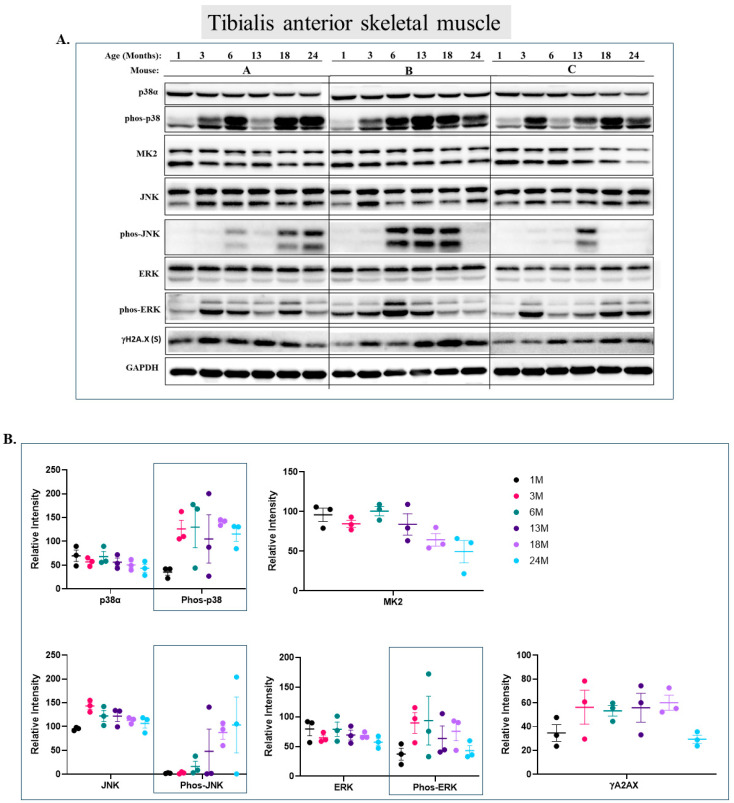
**Phosphorylation levels of p38α, ERKs, and JNK are essentially constant in the tibialis anterior during aging.** (**A**) Western blot analysis of protein lysates prepared from the tibialis anterior of wild-type mice, collected at the indicated age and tested with the indicated antibodies. Three mice were sacrificed at each age and were marked A, B, and C. (**B**) Relative intensity of Western blot results presented in panel A, as measured by ImageJ and normalized to the levels of GAPDH. Statistical analysis revealed no significant differences.

**Figure 6 ijms-25-01713-f006:**
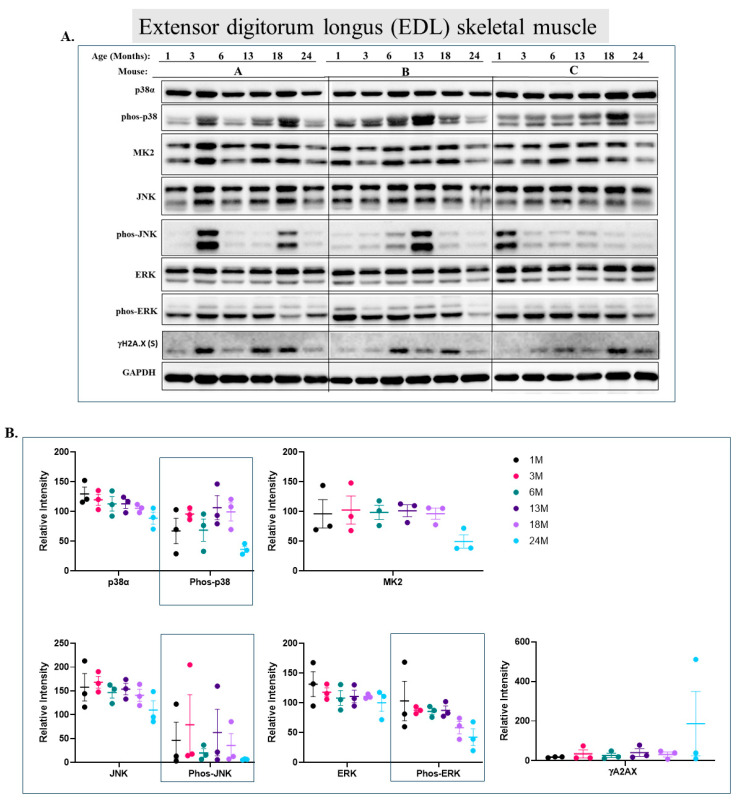
**Phosphorylation levels of p38α and ERKs are essentially constant in the EDL during aging.** (**A**) Western blot analysis of protein lysates prepared from the EDL of wild-type mice, collected at the indicated age and tested with the indicated antibodies. Three mice were sacrificed at each age and were marked A, B, and C. (**B**) Relative intensity of Western blot results presented in panel A, as measured by ImageJ and normalized to the levels of GAPDH. Statistical analysis revealed no significant differences.

**Figure 7 ijms-25-01713-f007:**
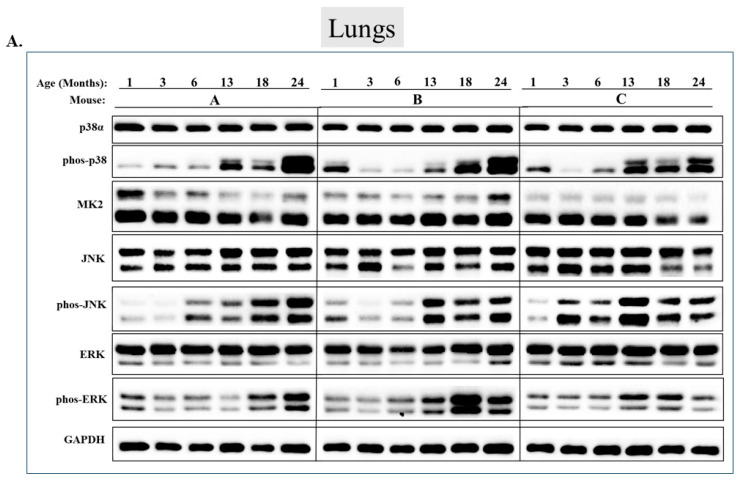
**Phosphorylation of p38α, JNK, and Erk1/2 is enhanced in the lungs during aging.** (**A**) Western blot analysis of protein lysates prepared from the lungs of wild-type mice, collected at the indicated age and tested with the indicated antibodies. Three mice were sacrificed at each age and were marked A, B, and C. (**B**) Relative intensity of Western blot results presented in panel A, as measured by ImageJ and normalized to the levels of GAPDH. Statistical significance of the differences of expression levels between 1- and 24-month-old and between 1- and 18-month-old mice was determined by an unpaired two-tailed *t*-test using GRAPHPAD PRISM software (v9.5.1) (ns = non-significant. Asterisks (*) mark *p*-value < 0.05).

**Figure 8 ijms-25-01713-f008:**
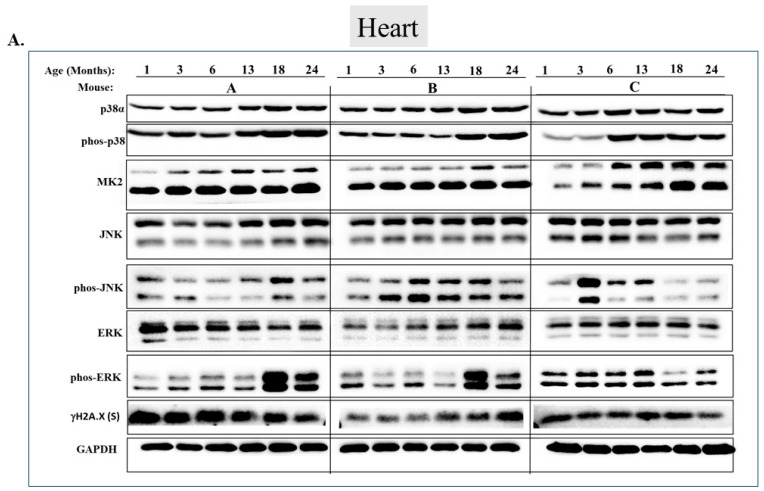
**Phosphorylation levels of p38α, JNK, and ERKs are essentially constant in the heart during aging.** (**A**) Western blot analysis of protein lysates prepared from the hearts of wild-type mice, collected at the indicated age and tested with the indicated antibodies. Three mice were sacrificed at each age and were marked A, B, and C. (**B**) Relative intensity of Western blot results presented in panel A, as measured by ImageJ then normalized to the levels of GAPDH. Statistical analysis revealed no significant differences.

**Figure 9 ijms-25-01713-f009:**
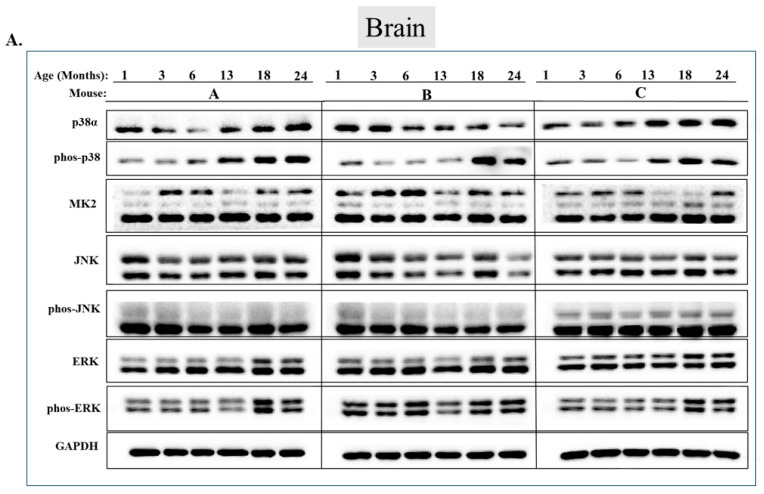
**Phosphorylation levels of p38α, JNK, and ERKs are essentially constant in the brain during aging.** (**A**) Western blot analysis of protein lysates prepared from the brains of wild-type mice, collected at the indicated age and tested with the indicated antibodies. Three mice were sacrificed at each age and were marked A, B, and C. (**B**) Relative intensity of Western blot results presented in panel A, as measured by ImageJ then normalized to the levels of GAPDH. Statistical analysis revealed no significant differences.

**Figure 10 ijms-25-01713-f010:**
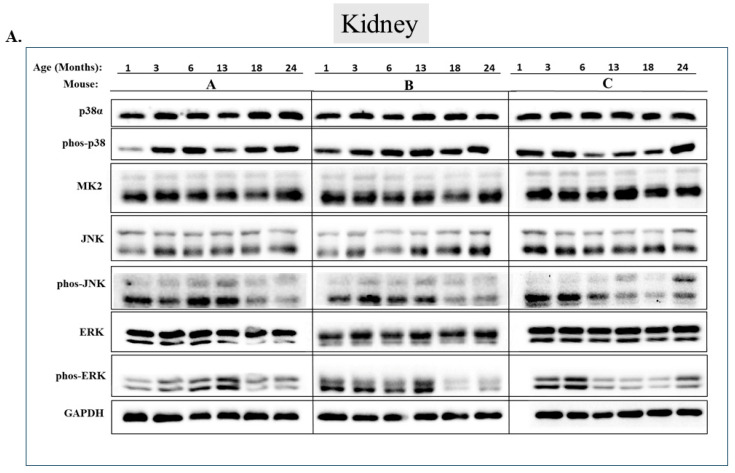
**Phosphorylation levels of p38α, JNK, and ERKs are essentially constant in the kidney during aging.** (**A**) Western blot analysis of protein lysates prepared from the kidneys of wild-type mice, collected at the indicated age and tested with the indicated antibodies. Three mice were sacrificed at each age and were marked A, B, and C. (Note: ERK and phos-ERK were probed on the same membrane with a stripping step). (**B**) Relative intensity of Western blot results presented in panel A, as measured by ImageJ then normalized to the levels of GAPDH. Statistical analysis revealed no significant differences.

**Figure 11 ijms-25-01713-f011:**
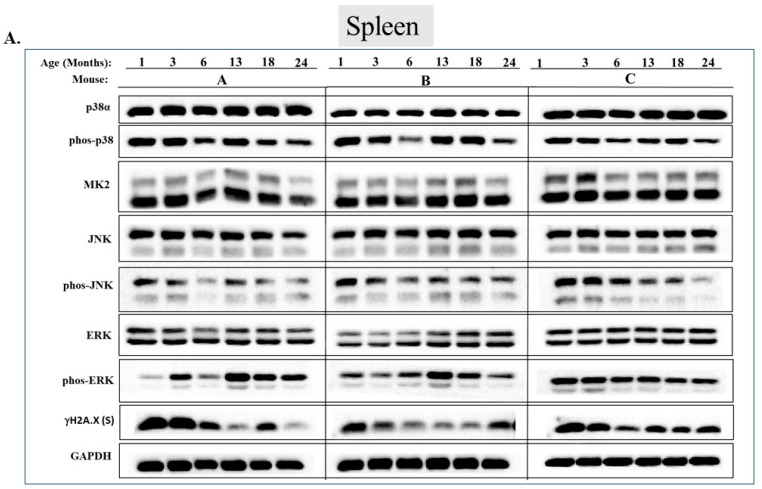
**Phosphorylation levels of p38α, JNK, and ERKs are essentially constant in the spleen during aging.** (**A**) Western blot analysis of protein lysates prepared from the spleens of wild-type mice, collected at the indicated age and tested with the indicated antibodies. Three mice were sacrificed at each age and were marked A, B, and C. (**B**) Relative intensity of Western blot results presented in panel A, as measured by ImageJ then normalized to the levels of GAPDH. Statistical analysis revealed no significant differences.

**Figure 12 ijms-25-01713-f012:**
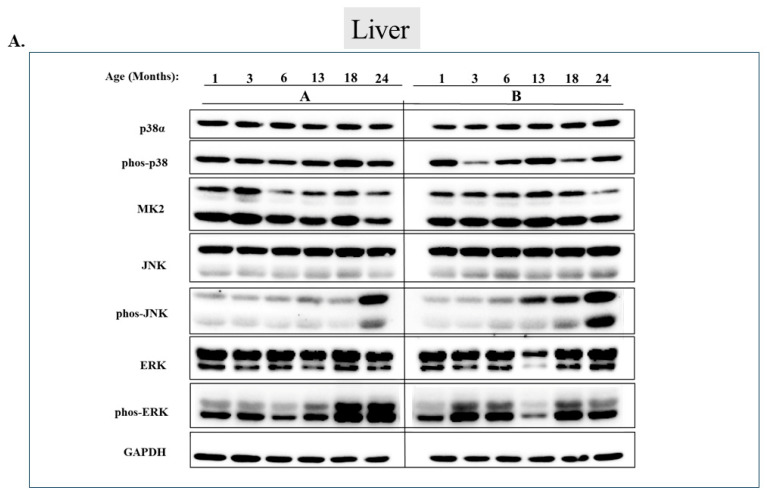
**Phosphorylation of JNK, but not of p38α or Erk1/2, is elevated in liver during aging.** (**A**) Western blot analysis of protein lysates prepared from the livers of wild-type mice, collected at the indicated age and tested with the indicated antibodies. Two mice were sacrificed at each age and were marked A and B. (**B**) Relative intensity of Western blot results presented in panel A, as measured by ImageJ then normalized to the levels of GAPDH. Statistical analysis revealed no significant differences.

**Figure 13 ijms-25-01713-f013:**
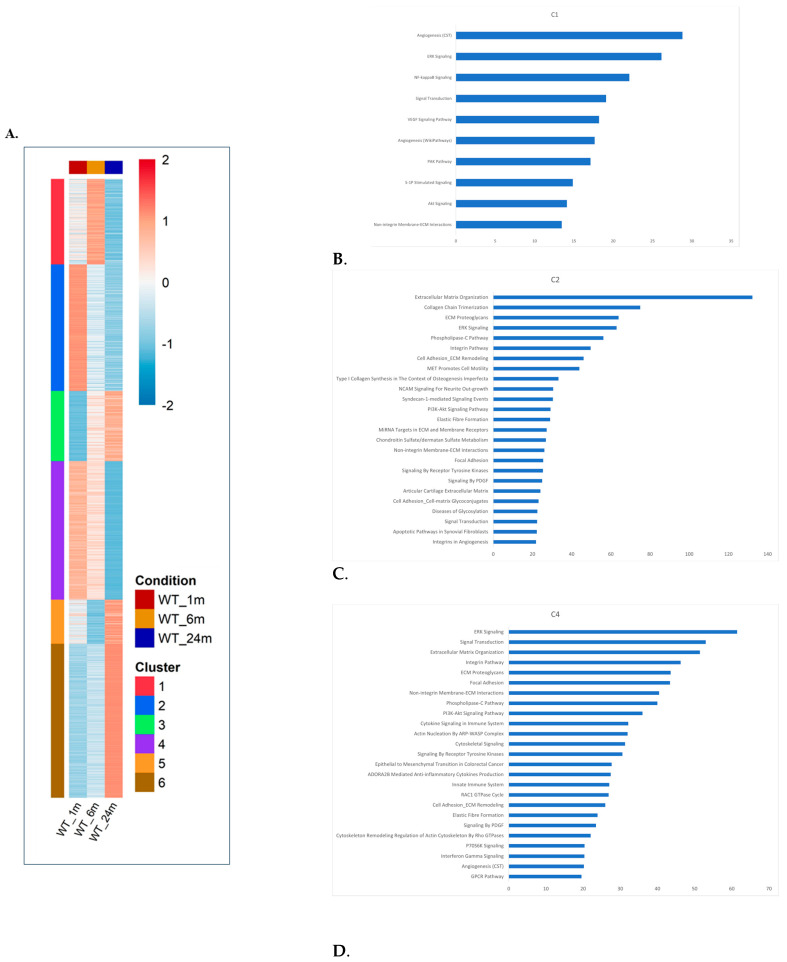
**Aging-dependent changes in mRNA expression levels of quadriceps muscle can be grouped into 6 clusters.** (**A**). Heat map of 6 different clusters into which affected mRNA molecules were grouped according to changes in expression kinetics for 1-, 6-, and 24-month-old mice. RNA was prepared from the quadriceps and analyzed by RNA-seq. Results at each time point are the average of 3 experimental repeats. (**B**–**F**). Enrichment analysis of genes in each of the 6 clusters shown in panel (**A**) using GeneAnalytics (Ben-Ari Fuchs et al. [62]). Enriched pathways with a high score (corrected *p*-value < 0.0001) are shown. The score is -log_2_(corrected *p*-value), i.e., the lower the bar, the lower the corrected *p*-value. A list of genes of each cluster is provided in Appendix A.

**Figure 14 ijms-25-01713-f014:**
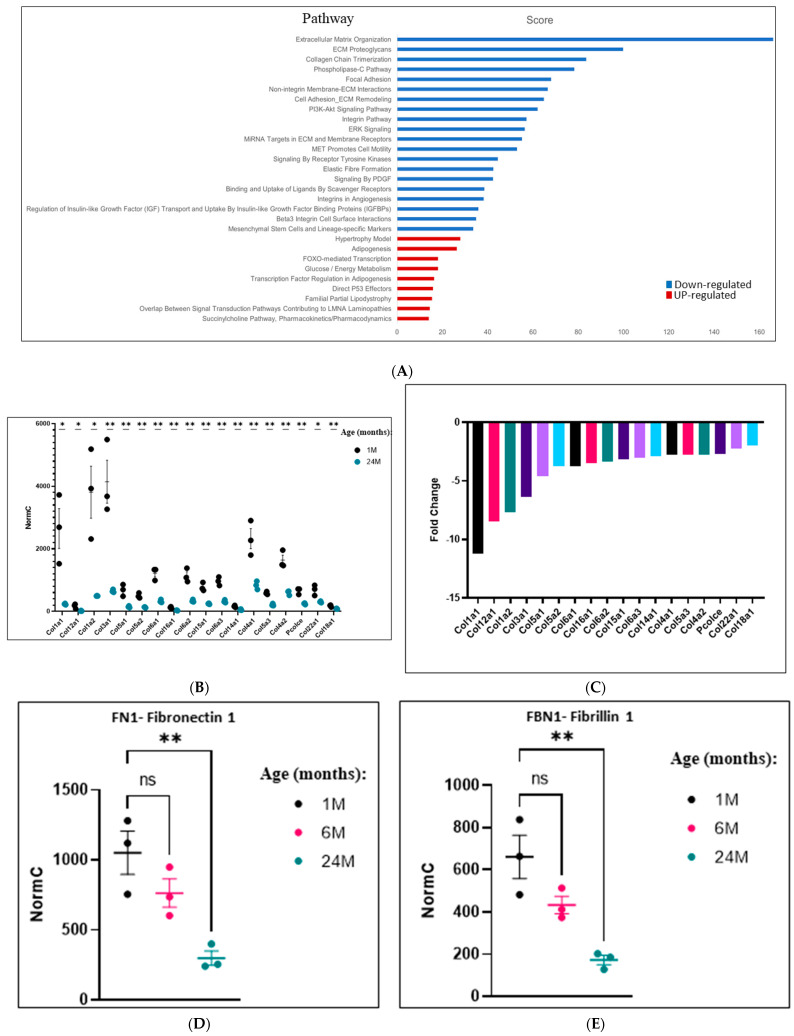
Analysis of differentially regulated genes revealed many groups of downregulated genes in aging, mostly related to ECM components. (**A**). Up and downregulated differentially expressed genes (DEGs) from the quadriceps of 1- vs. 24-month-old mice were analyzed using GeneAnalytics (Ben-Ari Fuchs et al. [62]). Enriched pathways with high score (corrected *p*-value < 0.0001) are shown. Pathways enriched in downregulated DEGs are colored blue, pathways enriched in upregulated DEGs are colored red. The score is −log_2_(corrected *p*-value), i.e., the lower the bar, the lower the corrected *p*-value. A list of genes is provided in Appendix A. (**B**). Normal counts of the mRNA molecules encoding the indicated collagens of quadriceps of 1- (black) and 24- (red) month-old mice, as revealed via RNA-seq analysis (results are the average and standard deviation of 3 mice that were tested at each time point). (**C**). Fold change (measures in 24-month divided by measures in 1-month-old mice) of the levels of the same mRNA molecules shown in panel (**A**,**D**,**E**). Normal counts of mRNA molecules encoding fibronectin 1 (FN (**C**) or fibrillin 1 (FBN1) (**D**) of quadriceps of wild-type mice aged 1, 6, and 24 months, as revealed via RNA-seq analysis (results shown are the average and standard deviation of 3 mice that were tested at each time point) (ns = non-significant. Asterisks (*) mark *p*-value < 0.05 (**) mark *p*-value < 0.01).

**Figure 15 ijms-25-01713-f015:**
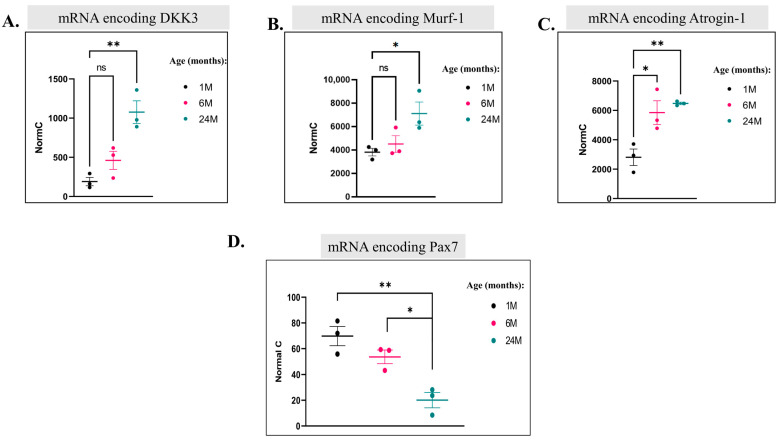
Aged skeletal muscle expresses higher levels of mRNAs encoding atrophy-associated proteins and Pax7 as compared to young mice. (**A**–**D**). Normalized counts of mRNA molecules encoding Dkk3 (**A**), Murf-1 (**B**), Atrogin-1 (**C**), and Pax7 (**D**) in the quadriceps of 1-, 6-, and 24-month-old mice, as revealed via RNA-seq analysis (results are the average and standard deviation of measures from 3 mice). (ns = non-significant. Asterisks (*) mark *p*-value < 0.05 (**) mark *p*-value < 0.01).

**Figure 16 ijms-25-01713-f016:**
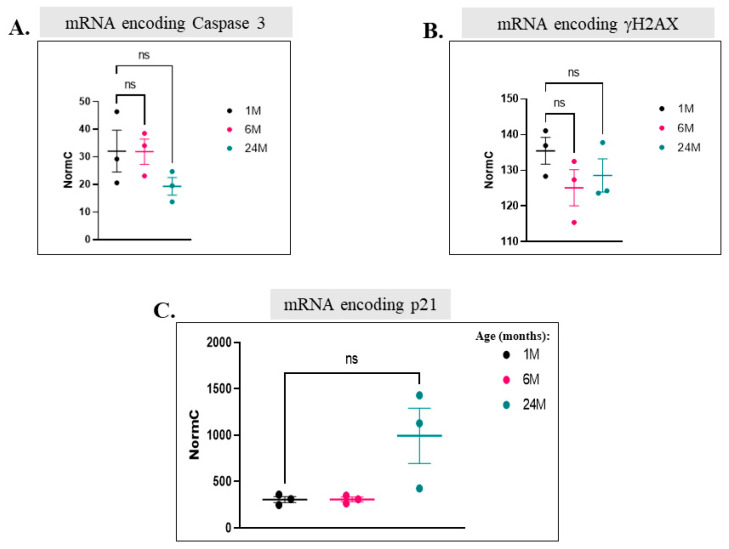
Aged skeletal muscle does not express higher levels than young mice of senescence-associated genes and genes encoding cell cycle inhibitors. (**A**–**C**). Normal counts of mRNA molecules encoding Caspase 3 (**A**), γH2AX1 (**B**), and p21 (**C**) in quadriceps of 1-, 6-, and 24-month-old mice, as revealed via RNA-seq analysis (results are the average and standard deviation of measures from 3 mice) (ns = non-significant).

## Data Availability

Data contained within the article.

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
