# Peer review of "Asynchronous Pattern of MAPKs’ Activity during Aging of Different Tissues and of Distinct Types of Skeletal Muscle"

_ijms, 2024, doi:10.3390/ijms25031713_

Round 1

Reviewer 1 Report

Comments and Suggestions for Authors

Asynchronous pattern of MAPKs activity during aging of different tissues and of distinct types of skeletal muscle

Review:This manuscript examined the changes in protein levels of the MAPK family in mice of different ages to explore whether it is related to aging, which is a meaningful study. But there are still some shortcomings that need to be modified and can be accepted.

1. Suggest adding the coresponding protein size to the edge of the protein imprint.

2. Suggest quantifying the results of protein imprinting, simply looking at images, and having so much protein data can easily confuse people.

3. Why are all the selected mice female?

4. Provide more detailed steps for protein blotting, as well as the concentrations of concentrated and separated gels.

5. N=3, is there too little common sample?

Line 344 Is there any evidence of membrane cutting on the H2A protein?

Line 344 Is there a slight error in the repeatability of the three protein samples? Sometimes, the trends of ERK, P-JNK, P-ERK, and H2A proteins are inconsistent at different time points.

Line 358 Why only ERK uses the stripping step?

Line 389 Why choose these three months for transcriptome testing?

Line 418 Missing a comma between age 1 6, and 24

Line 446 Please confirm the accuracy of the data. Is there still no significant difference in the sample results?

Comments on the Quality of English Language

It would be even better if writing could be more engaging, as it is too direct in describing the results and lacks continuity between the past and the future

Author Response

The reviewer suggests to add the corresponding protein size to the edge of the protein imprint.

We do understand the logic behind this request, but feel that more numbers and signs would make the anyway crowded figures impossible to look at. Also, each of the proteins tested in this study ran in the gel exactly at the expected apparent molecular weight and appeared as one single band (all original full-size gels are provided to the publisher office as requested). However, respecting the reviewer’s request and complying with the reviewers’ suggestion, in the revised version we added a statement to the materials and methods (under 4.4. Immunoblotting: All detected proteins appeared at the expected apparent molecular weight. All bands that appeared in the western blot assays are shown in the relevant figures. (lines 177 to 178).

The reviewer suggests quantifying the results of protein imprinting

We fully accept this suggestion. In the revised version levels of all proteins, as detected in the western blot assays, are presented as bars in which the measure in any single mouse is shown (Figures 1B, 3B, 4B, 5B, 6B, 7B, 8B, 9B, 10B, 11B, 12B).

The reviewer wonders why all the selected mice are female

The reason is simply not to mix genders. There are no reports suggesting that MAPKs behave differently in male or female mice.

The reviewer requests more detailed steps for protein blotting, as well as the concentrations of concentrated and separated gels

This information is provided under Materials and Methods, Immunoblotting (lines 169 to 177): Thirty μg of proteins' samples were separated on 12% SDS/PAGE (stacking gel concentration was 5%) and transferred to a nitrocellulose membrane (Bio-Rad) using Trans-Blot Turbo System (Bio-Rad). Membranes were blocked with 5% non-fat milk in TBST (1x PBS with 0.1%Tween) for 60 min. and then incubated with a primary antibody (diluted in 5% BSA in TBST) for 15 hours, washed and incubated with a HRP-conjugated secondary antibody (diluted in 5% nonfat dry milk in TBST) for 90 min. Signal was developed using Western BrightTM ECL Kit (K-12045-D50 Advantsa) or Maximum Sensitivity Substrate Kit (34096 Thermo Fisher). Signal was detected using ChemiDOCTM MS Imaging System instrument (Bio-Rad Laboratories, Hercules, CA, USA).

The reviewer says: N=3, is there too little common sample?

We agree with the reviewer that the more mice at each time point the better. Although 3 mice per time point is the minimum required to draw some statistic, it should be appreciated that we had 6 time points, so that altogether every tissue was tested in 18 mice. In many ‘aging’ studies the comparison is between very old and very young mice. We preferred to look at the kinetic as well.

The reviewer further comments:

Line 344 Is there any evidence of membrane cutting on the H2A protein? – the comment is not clear to us. We stress that no gel manipulation was performed whatsoever. Every gel, including those showing H2A are shown in full with no cutting. As said above antibodies are very specific and no non-specific bands appeared.

Line 344 Is there a slight error in the repeatability of the three protein samples? Sometimes, the trends of ERK, P-JNK, P-ERK, and H2A proteins are inconsistent at different time points. – The reviewer’s comment is correct and we fully accept it. This is in fact one of the central claims of our study; unlike previous reports that MAPKs, mainly p38a, are associated with aging in many tissues we claim that in fact they are not; and that in some tissues their activity is not consistent between different mice. This point, which is very clear particularly for JNK, is raised several time in the course of the study. 

Line 358 Why only ERK uses the stripping step? – The answer to this query is simple. As a rule

we refrain of performing any stripping and re-probing. In this study we were able, throughout the entire study, to run a specific gel and specific western blot assay for each tissue and antibody tested. Only in this one single case we did not have enough protein lysate, so for a one single case that we used a stripping step and explicitly mentioned it. We appreciate the reviewer for noting it.

Line 389 Why choose these three months for transcriptome testing? – As explained in detail in response to reviewer 3, this test is now removed from the report.

Line 418 Missing a comma between age 1 6, and 24 – we thanks the reviewer for noticing that. A comma is now added.

Line 446 Please confirm the accuracy of the data. Is there still no significant difference in the sample results? – this data is now removed from the paper.

Reviewer 2 Report

Comments and Suggestions for Authors

The authors Gilad et al., reported the asynchronous patten of MAPSs activity during aging of different tissues and of distinct types of skeletal muscle. They firstly tested the protein level of p-p38 and reported the phosphorylation elevated with aging only in the quadricep but not any other types of muscle. Also they looked at the p38alfa, JNK and Erk activity in lung, liver, heart and brain. They concluded that the quadricep was the only muscle that had increased p38 alfa phosphorylation and continued investigating the other potential gene expression that could be induced by p38 alfa phosphorylation. Although the manuscript provided a lot of information but it did not seem to be complete and figures were not compiled together. A few of the significant points that authors need to pay attention to are:

1) The mice that were used to measure p39alfa activity was not the same for different muscles. The quadriceps which they reported to have the most significant change only had one mouse western blotting shown. The other muscles had 3 mice. And it was clear that the western blotting results varied a lot from mouse to mouse. Therefore, the conclusion on quadricep is not supported by enough evidence. Also there was no quantification of the western.

2) n number was very small (n=2) when they analyzed the RNA seq data for Pax7-cre/p38alfa mice.

3) there seems to be a relationship between muscle fiber type and the p38alfa phosphorylation. However the authors did not touch on that.

Author Response

Reviewer 2

The reviewers claims that the mice that were used to measure p38alfa activity was not the same for different muscles. – We should stress that this is certainly not the case. Genetic backgrounds of all mice in the study were C57BL/6J. In any case, following the comment of reviewer 3 data with p38aMuscle mice was removed from the paper.

The reviewer says: the quadriceps which they reported to have the most significant change only had one mouse western blotting shown. The other muscles had 3 mice. And it was clear that the western blotting results varied a lot from mouse to mouse. Therefore, the conclusion on quadricep is not supported by enough evidence. Also there was no quantification of the western. We fully accept the comment and show now results of quadriceps from the additional 2 mice that show the same activation pattern (see results from 3 mice (marked A, B, C) in the new Fig. 3). We also show quantitation of the data (Figures 1B, 3B, 4B, 5B, 6B, 7B, 8B, 9B, 10B, 11B, 12B).

Reviewer: n number was very small (n=2) when they analyzed the RNA seq data for Pax7-cre/p38alfa mice. – we agree: data with these mice is removed from the paper.

Reviewer: there seems to be a relationship between muscle fiber type and the p38alfa phosphorylation. However the authors did not touch on that. – This comment is not fully understood. We do not see in fact any relationships between muscle type and p38a phosphorylation. Specifically, QC and GC were both mix of fast and slow fibers and the activity pattern of MAPKs is different

Reviewer 3 Report

Comments and Suggestions for Authors

Dear authors,

First of all, I would like to thank authors for submitting this work.

In this work, authors investigated the p38alpha signaling in various tissues (including various types of muscle) of mice. This study is mainly aimed to understand how p38 (& ERK) signaling is regulated in the muscle tissues of aging mice therefor may shed light on the biology of muscle aging. Authors mainly performed a series of western blotting and gene expression analysis by RNASeq.

One of the major shortcomings of this manuscript is the lack of quantitation for the wester blots. The first 12 figures consists of western blots of multiple proteins in various tissues. I strongly advice authors to present these results as a bar chart (with each mouse presented as a dot). This will allow the readers to follow the paper easier. As its current shape, it is very hard for me to analyze the first 12 figures. And therefore, cannot really assess the big portion of the paper.

I think the RNASeq analysis on 1, 6 and 24 months quadriceps muscle is a great addition to this manuscript. Can you please perform (& present) deeper analysis on each of the 6 clusters? I expected a figure after figure 13 where these data would be presented. Therefore, I suggest authors to perform IPA analysis on each of these 6 clusters and present the most enriched pathways as a new figure.

The data starting from Figure 15 (until 21) is using a mouse model named p38.D176A+F327S. However the reference is only a manuscript ‘submitted’ not published.  Therefore, I have some concerns on these data as a whole.

In any case, for Figures 14-21: can you please convert the bar plots showing individual data points (dot plots)?

Finally, I would like to make a suggesting on order of the figures. For examples, I suggest authors to considering presenting Figure 16 earlier.

In summary, I will be happy to re-assess this paper once the above changes and additions are made.

Comments on the Quality of English Language

Moderate English language revision can be necessary.

Author Response

The reviewer says: One of the major shortcomings of this manuscript is the lack of quantitation for the wester blots. The first 12 figures consists of western blots of multiple proteins in various tissues. I strongly advice authors to present these results as a bar chart – As explained in our response to the first comment of reviewer 1, we fully accept the need for quantitation and thanks reviewers 1 and 3 for suggesting it. In the revised version, levels of all proteins, as detected in the western blot assays, were quantified and presented as bars (Figures 1B, 3B, 4B, 5B, 6B, 7B, 8B, 9B, 10B, 11B, 12B).

The reviewer further says: I think the RNASeq analysis on 1, 6 and 24 months quadriceps muscle is a great addition to this manuscript. Can you please perform (& present) deeper analysis on each of the 6 clusters? I expected a figure after figure 13 where these data would be presented. Therefore, I suggest authors to perform IPA analysis on each of these 6 clusters and present the most enriched pathways as a new figure. Here too, we fully accepted the reviewer’s suggestion, performed analysis on all clusters and present the results now as Figs. 13B-G.

Reviewer: The data starting from Figure 15 (until 21) is using a mouse model named p38.D176A+F327S. However the reference is only a manuscript ‘submitted’ not published.  Therefore, I have some concerns on these data as a whole. – Given the concerns raised by the reviewer regarding the data, we removed it altogether.

Reviewer: In any case, for Figures 14-21: can you please convert the bar plots showing individual data points (dot plots)? – Data was converted to dot plots

Reviewer: Finally, I would like to make a suggesting on order of the figures. For examples, I suggest authors to considering presenting Figure 16 earlier. Comment accepted. In the revised version, old figure 16 is Fig. 14A.

Round 2

Reviewer 3 Report

Comments and Suggestions for Authors

Dear authors,

Thanks for responding my comments and revising the manuscript. I suggest to perform statistical analysis on the quantifications performed on western blots.

Comments on the Quality of English Language

Minor English language editing might be necessary.

Author Response

The reviewer suggest to perform statistical analysis on the quantifications performed on western blots.

We added the results of our statistical analysis in figures where at least one of the MAPK have significant differences between 1 vs. 24 months or 1 vs. 18 months (unpaired two-tailed t-test (GRAPHPAD PRISM software) (*P < 0.05, **P < 0.01, ***P < 0.001, ****P < 0.0001).